## Effect of COVID-19 lockdown on hospital admissions and mortality in rural KwaZulu-Natal, South Africa: interrupted time series analysis

Amy McIntosh [1], Max Bachmann [1], Mark J Siedner [2,3]
Dickman Gareta [2], Janet Seeley [2,4], Kobus Herbst [2,5]

► Prepublication history and additional materials for this paper are available online. To view these files, please visit the journal online (http://dx.doi.org/10.1136/bmjopen-2020-047961).

¹Faculty of Medicine and Health Sciences, University of East Anglia, Norwich, UK
²Africa Health Research Institute, KwaZulu-Natal, South Africa
³Department of Medicine, Harvard Medical School, Boston, Massachusetts, USA
⁴Department of Global Health and Development, London School of Hygiene and Tropical Medicine, London, UK
⁵DSI-MRC South African Population Research Infrastructure Network, Durban, South Africa

**Correspondence to**
Dr Amy McIntosh;
A.Mcintosh@doctors.org.uk

## ABSTRACT

**Objective**  To assess the effect of lockdown during the 2020 COVID-19 pandemic on daily all-cause admissions, and by age and diagnosis subgroups, and the odds of all-cause mortality in a hospital in rural KwaZulu-Natal (KZN).

**Design**  Observational cohort.

**Setting**  Referral hospital for 17 primary care clinics in uMkhanyakude District.

**Participants**  Data collected by the Africa Health Research Institute on all admissions from 1 January to 20 October: 5848 patients contributed to 6173 admissions.

**Exposure**  Five levels of national lockdown in South Africa from 27 March 2020, with restrictions decreasing from levels 5 to 1, respectively.

**Outcome measures**  Changes and trends in daily all-cause admissions and risk of in-hospital mortality before and at each stage of lockdown, estimated by Poisson and logistic interrupted time series regression, with stratification for age, sex and diagnosis.

**Results**  Daily admissions decreased during level 5 lockdown for infants (incidence rate ratio (IRR) compared with prelockdown 0.63, 95% CI 0.44 to 0.90), children aged 1–5 years old (IRR 0.43, 95% CI 028 to 0.65) and respiratory diagnoses (IRR 0.57, 95% CI 0.36 to 0.90). From level 4 to level 3, total admissions increased (IRR 1.17, 95% CI 1.06 to 1.28), as well as for men >19 years (IRR 1.50, 95% CI 1.17 to 1.92) and respiratory diagnoses (IRR 4.26, 95% CI 2.36 to 7.70). Among patients admitted to hospital, the odds of death decreased during level 5 compared with prelockdown (adjusted OR 0.48, 95% CI 0.28 to 0.83) and then increased in later stages.

**Conclusions**  Level 5 lockdown is likely to have prevented the most vulnerable population, children under 5 years and those more severely ill from accessing hospital care in rural KZN, as reflected by the drop in admissions and odds of mortality. Subsequent increases in admissions and in odds of death in the hospital could be due to improved and delayed access to hospital as restrictions were eased.

## BACKGROUND

The COVID-19 global pandemic has caused huge disruption to health systems worldwide. As the number of cases of the novel coronavirus rose throughout 2020, health systems focused resources on to COVID-19.

### Strengths and limitations of this study

► Interrupted time series regression provides original insight into the impact of COVID-19 lockdown on hospital admissions and mortality in South Africa.

► This paper complements recently published evidence from the same catchment to give a fuller picture of access to multiple levels of healthcare in one population.

► Hospital data provide only indirect evidence of access to hospital care and reasons for changes in access.

► Using data from one hospital in a rural, low-middle-income setting in South Africa limits generalisability to other settings.

► Small sample limits statistical power to detect changes in subgroups.

This led to global reductions in resources for non-COVID-19 services.[1–4] Concurrently, governments attempted to reduce community transmission with lockdowns, restricting movement and activities of populations. Such restrictions have had far-reaching impact on all aspects of society, including economies, education, unemployment, inequalities and health.[5–7]

The effects of these measures on progress to reach universal health coverage globally are largely unknown, but it is likely that progress in low-income and middle-income countries was adversely affected due to weak and under-resourced health systems serving populations with high burden of both communicable and non-communicable diseases, as well as social and economic deprivation.[8] Concerns have been voiced about the impact on HIV care, family planning services, and maternal and child health.[9–11]

A potential consequence of lockdowns is reduced utilisation for non-COVID-19 conditions. Excess mortality during the pandemic

| Level 5 | Level 4 | Level 3 | Level 2 | Level 1 |
|---|---|---|---|---|
| 26th March – 30th April 2020 | 1st – 31st May 2020 | 1st June – 17th August 2020 | 18th August – 20th September 2020 | 21st September 2020 onwards |
| • Essential travel only <br> • Businesses closed <br> • Schools closed <br> • Transport restrictions <br> • Alcohol ban | • Some workplaces reopen <br> • Outdoor exercise permitted | • All workplaces & schools reopen <br> • Public transport restarts <br> • Physical distancing <br> • Alcohol ban lifted temporarily until 12th July | • Small gatherings of <50 people <br> • Leisure and social activities permitted <br> • Alcohol ban lifted | • Minimal restrictions <br> • All activities to be physically distanced and mask-wearing |

**Figure 1**  Levels of national COVID-19 lockdown during 2020 in South Africa.

that is not explained by direct mortality from COVID-19 supports this concern.[12 13] Predicted effects, both through reduced health service provision and reduced utilisation, include higher rates of mortality from vaccine-preventable diseases, HIV and tuberculosis in children aged under 5 years and pregnant women.[14–17]

South Africa implemented a level 5 lockdown order at midnight of 26 March 2020.[18] This prohibited non-essential movement and closure of non-essential businesses, schools and services. Health workers were exempt from movement restrictions, as was accessing health services. However, this led to significant reductions in availability of transport and reductions in income, particularly for informal or non-essential workers. Restrictions were eased incrementally over time (figure 1).[19–21] The lowest level of lockdown, level 1, was initiated on 21 September[22] and continued beyond the end of this study (20 October 2020).

Lockdown restrictions applied nationwide, including the KwaZulu-Natal province, despite COVID-19 cases and deaths only increasing after the end of May[23] and peaking around 25 July. COVID-19 deaths peaked around 8 August. By 20 October daily cases and deaths in the province had decreased to less than 10 and 1 per million population, respectively. The effects of levels 5, 4 and 3 lockdown on the utilisation of primary care facilities in this mostly rural KwaZulu-Natal subdistrict have been reported by Siedner and colleagues.[24] They found that level 5 lockdown was associated with a significant reduction in visits to primary care clinics for child health and in those under 5 years old, but not for total visits or for adult care.

In this study we investigated the effects of the lockdown measures during the COVID-19 pandemic on admissions to and deaths in the referral hospital serving the same clinics and population as the latter study. We focused particularly on admissions in subgroups stratified by age, sex and diagnosis as indicators of access to hospital care. We also compared the risk of in-hospital mortality before and after lockdown as an indicator of the severity of presenting illnesses. We hypothesised that lockdown measures would lead to reduced hospital admissions due to reduced mobility and would increase as lockdown restrictions were eased. We also hypothesised that risk

of in-hospital mortality would increase if people with less severe illness were selectively deterred from going to hospitals or would decrease if those with more severe illness were selectively deterred.

## METHODS

### Study population

We performed secondary analysis of data collected by the Africa Health Research Institute (AHRI), located in a subdistrict of northern KwaZulu-Natal, uMkhanyakude District in South Africa. It is a largely rural district which is socioeconomically deprived, falling in the bottom 10 of all districts in South Africa in terms of education levels, employment, access to piped water and sanitation at home.[25] The study population was patients admitted to Hlabisa Hospital from 1 January 2020 to 20 October 2020. It is the main referral hospital for 17 clinics in the subdistrict and is in an urban centre. Only 5% of the population hold medical insurance and so are mostly dependent on the hospital and clinics for their healthcare.[25]

### Data collection

Data for each admission were captured prospectively by AHRI clinical research workers based in the hospital. These include demographic data of each patient, as well as admission, discharge and death diagnoses, coded according to the International Classification of Diseases (ICD-10) by trained nurses and recorded on an electronic database.[26] Women admitted for elective caesarean section, antenatal care or other intrapartum care were included, excluding those admitted for normal vaginal deliveries.

### Statistical analysis

Total admissions for the study period were stratified by age, sex and discharge diagnosis to demonstrate the typical case-mix at the hospital. To identify suitable regression models for estimating the effects of lockdown, we first graphically plotted the number of daily admissions and probability of death over time with locally weighted scatterplot smoothing (LOWESS).

We used interrupted time series linear regression to estimate the effects of lockdown on the number of daily admissions to hospital. We first calculated the number of admissions each day then analysed the aggregated data at day level. These analyses were carried out for all admissions and for subgroups defined by age, sex and diagnosis. We used two sets of regression models to estimate the effect of lockdown on hospital utilisation, one estimating step changes in daily admissions or odds of death during each level of lockdown compared with prelockdown, and the other estimating linear time trends over the entire lockdown period.

To estimate absolute changes in the mean number of daily admissions between each level of lockdown, compared with the prelockdown period, we used the following model:

$$Yt = \beta_0 + \beta_1 L_5 + \beta_2 L_4 + \beta_3 L_3 + \beta_4 L_2 + \beta_5 L_1 + \beta_{6-11} Z$$

$Yt$ is the daily number of hospital admissions. $\beta_0$ is the mean daily number of visits or admissions immediately before lockdown (the regression intercept). $L_5$, $L_4$, $L_3$, $L_2$ and $L_1$ are binary dummy variables indicating each stage of lockdown, and their coefficients ($\beta_1 - \beta_5$) are mean differences from the prelockdown period; differences between successive levels were the differences between the corresponding coefficients. $Z$ is a vector of dummy variables representing day of week.

The regression equation to estimate the continuous time trend before and after level 5 lockdown and the step change at the start of level 5 was as follows:

$$Yt = \beta_0 + \beta_1 T + \beta_2 X_t + \beta_3 TX_t + \beta_4 Z$$

$Yt$, $\beta_0$ and $Z$ are the same as in the previous equation. $\beta_1$ is the weekly change in daily admissions before lockdown (ie, coefficient of the slope before lockdown) and $T$ represents the number of weeks before or after lockdown. $\beta_2$ is the step change in number of admissions immediately after level 5 lockdown on 27 March 2020. $X_t$ represents the prelockdown and postlockdown periods coded as a binary variable. $\beta_3$ is the change in trend after lockdown. $TX_t$ is the time–lockdown interaction term; $\beta_1 + \beta_3$ is thus the time trend after lockdown. We used the results of each analysis to graphically plot predicted continuous time trends and step changes.

We used Poisson regression to estimate relative changes and then used linear regression to estimate absolute changes in daily admissions associated with lockdown for both models. We plotted frequency distributions of residuals of linear regression models, overall and in subgroups, to assess their normality. Predicted values from the linear regression models with continuous time trends were plotted against the number of weeks before and after lockdown.

The odds of in-hospital mortality among admitted patients were investigated using equivalent logistic regression models (with $Yt$ representing log odds of death). These analyses were at the level of individual admission, with robust adjustment for intraperson correlation of outcomes during repeated admissions. Age, sex and diagnosis (ICD-10 chapter) were added as covariates instead of carrying out separate analyses for each subgroup. To graphically plot changes in probability of death before and after the start of level 5 lockdown, we calculated the probability of death each week using a linear regression model equivalent to the second equation.

As a sensitivity analysis, to assess the effect of seasonal confounding, we extended the study period to the start of 2019 and added calendar month as covariates to each model.

We use a 5% significance level and 95% CI. Analysis was carried out with Stata V.16.0.

## Patient and public involvement

There was no patient or public involvement in this study.

## RESULTS

There were 6173 admissions by 5848 individuals to the study hospital from 1 January 2020 to 20 October 2020 (table 1). Of the admissions 4.7% (n=291) resulted in death. Of all admissions, 59.4% were aged 20–45 years old (n=3664), and more admissions were among females compared with males (77.6% vs 22.4%, n=4788 vs n=1385). The most common reasons for admission were maternal or neonatal conditions (32.0%, n=3271), communicable diseases (CDs) (21.6%, n=2203) and non-communicable diseases (NCDs) (14.2%, n=1455). Ninety-two admissions were identified as having COVID-19, of which 66.3% (n=61) occurred during level 3 lockdown and 21.7% (n=20) during level 2 lockdown. LOWESS plots of crude time trends in admissions and mortality during 2019 and 2020 are shown graphically in online supplemental figures 1–3.

The incidence rate ratios (IRRs) of daily admissions between each level of lockdown are shown in table 2. All-cause daily admissions did not change significantly between levels, although they increased slightly from levels 4 to 3 (IRR 1.17, 95% CI 1.06 to 1.28).

However, there were statistically significant changes among subgroups of patients. Daily admissions decreased significantly during level 5 lockdown for infants (IRR 0.63, 95% CI 0.44 to 0.90), children aged 1–5 years old (IRR 0.43, 95% CI 028 to 0.65) and respiratory diagnoses (IRR 0.57, 95% CI 0.36 to 0.90). In infants this represents a reduction in the incidence rate from 1.46 to 0.91 admissions per day, which did not recover by the end of the study, with 0.42 admissions per day during level 1 lockdown. The same trend was seen in children aged 1–5 years old, with an incidence rate before lockdown of 1.07 per day, reducing to 0.43 admissions per day during level 5, and not recovering by level 1 with an incidence rate of 0.61 per day. The most common reasons for admission in infants were neonatal conditions (42.6%), gastroenteritis (18.6%) and pneumonia (9.5%). In children aged 1–5 years old they were injuries (32.1%), gastroenteritis (15.0%) and pneumonia (10.6%).

**Table 1** Number (%) of admissions to and deaths in Hlabisa Hospital in rural KwaZulu-Natal, South Africa, from January 2020 to September 2020, according to sex, age and diagnosis

| | Total, n (% of column total) | Male, n (% of row total) | Female, n (% of row total) | Age <1, n (% of row total) | Age 1–5, n (% of row total) | Age 6–19, n (% of row total) | Age 20–45, n (% of row total) | Age 46–65, n (% of row total) | Age >65, n (% of row total) |
|---|---|---|---|---|---|---|---|---|---|
| Total, n (%) | 6173 (100.0)* | 1385 (22.4) | 4788 (77.6) | 334 (5.4) | 242 (3.9) | 951 (15.4) | 3664 (59.4) | 580 (9.4) | 402 (6.5) |
| Maternal and neonatal, n (%) | 3271 (32.0) | 78 (2.4) | 3193 (97.6) | 145 (4.4) | 7 (0.2) | 662 (20.2) | 2446 (74.8) | 6 (0.2) | 5 (0.2) |
| Non-communicable diseases, n (%) | 1455 (14.2) | 568 (39.0) | 887 (61.0) | 59 (4.1) | 70 (4.8) | 109 (7.5) | 517 (35.5) | 386 (26.5) | 314 (21.6) |
| Communicable diseases, n (%) | 2203 (21.6) | 591 (26.8) | 1612 (73.2) | 129 (5.9) | 103 (4.7) | 110 (5.0) | 1448 (65.7) | 280 (12.7) | 133 (6.04) |
| Injuries, n (%) | 496 (4.9) | 302 (60.9) | 194 (39.1) | 19 (3.8) | 83 (16.7) | 104 (21.0) | 225 (45.4) | 46 (9.3) | 19 (3.8) |
| Respiratory, n (%) | 323 (3.2) | 151 (46.8) | 172 (53.3) | 42 (13.0) | 34 (10.5) | 15 (4.6) | 92 (28.5) | 69 (21.4) | 71 (22.0) |
| Other, n (%) | 2786 (27.3) | 556 (20.0) | 2230 (80.0) | 194 (7.0) | 124 (4.5) | 672 (35.5) | 1531 (55.0) | 158 (5.7) | 107 (3.8) |
| Deaths, n (%) | 291 (4.7) | 139 (47.8) | 152 (52.2) | 18 (6.2) | 2 (0.7) | 7 (2.4) | 111 (38.14) | 80 (27.5) | 73 (25.1) |

*Primary and secondary diagnoses used for each admission and therefore total of column % exceeds 100%.

From level 4 to level 3, admissions increased significantly for men aged over 19 (IRR 1.50, 95% CI 1.17 to 1.92), individuals 20–45 years old (IRR 1.19, 95% CI 1.05 to 1.34) and respiratory diagnoses (IRR 4.26, 95% CI 2.36 to 7.70).

From level 2 to level 1, admissions decreased significantly for maternal and neonatal diagnoses (IRR 0.74, 95% CI 0.64 to 0.86), NCDs (IRR 0.76, 95% CI 0.60 to 0.96) and injuries (IRR 0.54, 95% CI 0.34 to 0.86). As a sensitivity analysis, admissions from 1 October onwards were excluded to test if this finding was due to data collection or upload delays in ICD-10 coding. The decrease seen in NCDs persisted but became non-significant (IRR 0.79, 95% CI 0.56 to 1.11). Maternal and neonatal admissions no longer decreased (IRR 1.00, 95% CI 0.83 to 1.22). Admissions for injuries continued to decrease (IRR 0.43, 95% CI 0.19 to 0.94). There were no statistically significant changes in these subgroups during the earlier stages of lockdown. Daily number of admissions in individuals aged 6–19, 45–65 and over 65 years old and for adult women and CDs did not change significantly at any stage of lockdown.

Linear regression models of these step changes produced similar results (online supplemental table 1). Residuals of linear regression models were approximately normally distributed. Poisson and linear regression models of continuous time trends before and after the start of level 5 lockdown produced similar results, although some differences became non-significant (online supplemental tables 2 and 3, figures 2 and 3).

Among patients admitted to hospital, the odds of death decreased substantially and significantly during level 5 lockdown compared with prelockdown (adjusted OR 0.48, 95% CI 0.28 to 0.83), then increased from level 5 to level 4 (OR 1.94, 95% CI 1.01 to 3.71), did not change from level 4 to level 3 (OR 0.84, 95% CI 0.51 to 1.39), increased from level 3 to level 2 (OR 1.74, 95% CI 1.15 to 2.63), and then did not change during level 1 (table 3). These changes were not affected by adjustment for age, sex or diagnosis, except that adjustment for diagnosis increased the ORs for levels 4 vs 5 and levels 2 vs 3 and made them statistically significant (table 3 and online supplemental table 4). Figure 4 illustrates the large decrease in probability of death immediately after the start of level 5 lockdown, which then increases over time.

Sensitivity analysis with adjustment for calendar month, including admissions from the start of 2019, did not substantially alter the direction of changes in daily admissions or effect size seen during lockdown. For example, admissions in children aged 1–5 still decreased significantly at level 5 lockdown (IRR 0.45, 95% CI 0.29 to 0.68) when data from 2019 were included in analysis. However, these results are not included as LOWESS plots identified an overall decrease in admissions since the start of 2019 and this may have exaggerated the effect size seen during lockdown.

**Table 2** Step changes in daily incidence of admissions at each level of lockdown: Poisson regression models adjusted for day of week

| | Prelockdown admission daily incidence rate (95% CI) | | IRR at level 5 vs prelockdown | 95% CI | P value | IRR at level 4 vs 5 | 95% CI | P value | IRR at level 3 vs 4 | 95% CI | P value | IRR at level 2 vs 3 | 95% CI | P value | IRR at level 1 vs 2 | 95% CI | P value |
|---|---|---|---|---|---|---|---|---|---|---|---|---|---|---|---|---|---|
| Total | 17.08 | (15.73 to 18.55) | 0.96 | 0.88 to 1.05 | 0.41 | 0.99 | 0.88 to 1.10 | 0.81 | 1.17* | 1.06 to 1.28 | 0.001 | 1.02 | 0.94 to 1.11 | 0.67 | 1.02 | 0.94 to 1.11 | 0.67 |
| Male >19 | 2.59 | (2.11 to 3.19) | 0.95 | 0.75 to 1.21 | 0.70 | 0.93 | 0.69 to 1.25 | 0.61 | 1.50* | 1.17 to 1.92 | 0.001 | 0.98 | 0.80 to 1.21 | 0.87 | 0.93 | 0.72 to 1.21 | 0.59 |
| Female >19 | 8.96 | (8.02 to 10.00) | 1.05 | 0.93 to 1.18 | 0.43 | 1.03 | 0.89 to 1.18 | 0.71 | 1.11 | 0.99 to 1.25 | 0.08 | 1.04 | 0.93 to 1.16 | 0.49 | 0.92 | 0.80 to 1.05 | 0.20 |
| Age <1 | 1.46 | (1.04 to 2.03) | 0.63* | 0.44 to 0.90 | 0.01 | 0.60 | 0.35 to 1.02 | 0.06 | 1.52 | 0.93 to 2.49 | 0.09 | 0.75 | 0.48 to 1.18 | 0.21 | 1.17 | 0.67 to 2.02 | 0.58 |
| Age 1–5 | 1.07 | (0.99 to 1.17) | 0.43* | 0.28 to 0.65 | <0.001* | 0.88 | 0.47 to 1.66 | 0.70 | 1.50 | 0.88 to 2.55 | 0.14 | 0.57* | 0.33 to 0.98 | 0.04 | 1.75 | 0.94 to 3.28 | 0.08 |
| Age 6–19 | 3.11 | (2.56 to 3.79) | 0.99 | 0.80 to 1.24 | 0.95 | 1.05 | 0.80 to 1.37 | 0.74 | 0.98 | 0.78 to 1.24 | 0.89 | 1.17 | 0.95 to 1.45 | 0.14 | 0.76 | 0.58 to 0.99 | 0.05 |
| Age 20–45 | 9.13 | (8.18 to 10.18) | 1.08 | 0.96 to 1.21 | 0.20 | 0.95 | 0.83 to 1.10 | 0.50 | 1.19* | 1.05 to 1.34 | 0.01 | 1.05 | 0.95 to 1.17 | 0.34 | 0.90 | 0.79 to 1.03 | 0.12 |
| Age 46–65 | 1.29 | (0.96 to 1.72) | 0.89 | 0.65 to 1.21 | 0.44 | 1.24 | 0.86 to 1.78 | 0.26 | 1.22 | 0.91 to 1.64 | 0.18 | 0.77 | 0.58 to 1.03 | 0.08 | 1.16 | 0.82 to 1.65 | 0.40 |
| Age >65 | 1.14 | (0.83 to 1.57) | 0.80 | 0.56 to 1.17 | 0.25 | 1.29 | 0.83 to 2.01 | 0.26 | 1.04 | 0.73 to 1.48 | 0.83 | 1.18 | 0.86 to 1.63 | 0.31 | 0.83 | 0.55 to 1.23 | 0.35 |
| Maternal and neonatal | 8.28 | (7.38 to 9.30) | 1.09 | 0.97 to 1.23 | 0.17 | 1.04 | 0.90 to 1.21 | 0.56 | 1.03 | 0.91 to 1.16 | 0.68 | 1.08 | 0.97 to 1.21 | 0.17 | 0.74 | 0.64 to 0.86 | <0.001 |
| NCDs | 3.84 | (3.23 to 4.57) | 0.90 | 0.75 to 1.08 | 0.27 | 1.15 | 0.92 to 1.43 | 0.23 | 1.07 | 0.89 to 1.29 | 0.46 | 0.90 | 0.76 to 1.08 | 0.27 | 0.76 | 0.60 to 0.96 | 0.02 |
| CDs | 6.34 | (5.51 to 7.29) | 0.88 | 0.77 to 1.02 | 0.10 | 0.95 | 0.79 to 1.15 | 0.62 | 1.14 | 0.98 to 1.34 | 0.10 | 0.91 | 0.78 to 1.05 | 0.20 | 0.86 | 0.71 to 1.04 | 0.11 |
| Injuries | 2.56 | (2.01 to 3.00) | 0.80 | 0.59 to 1.08 | 0.14 | 0.83 | 0.56 to 1.24 | 0.37 | 1.34 | 0.95 to 1.88 | 0.10 | 0.88 | 0.64 to 1.20 | 0.42 | 0.54 | 0.34 to 0.86 | 0.01 |
| Respiratory | 0.70 | (0.47 to 1.06) | 0.57 | 0.36 to 0.90 | 0.02 | 0.59 | 0.30 to 1.20 | 0.15 | 4.26* | 2.36 to 7.70 | <0.001 | 0.58* | 0.40 to 0.85 | 0.01 | 0.89 | 0.53 to 1.48 | 0.65 |

*Statistically significantly different from prelockdown.
CDs, communicable diseases; IRR, incidence rate ratio; NCDs, non-communicable diseases.

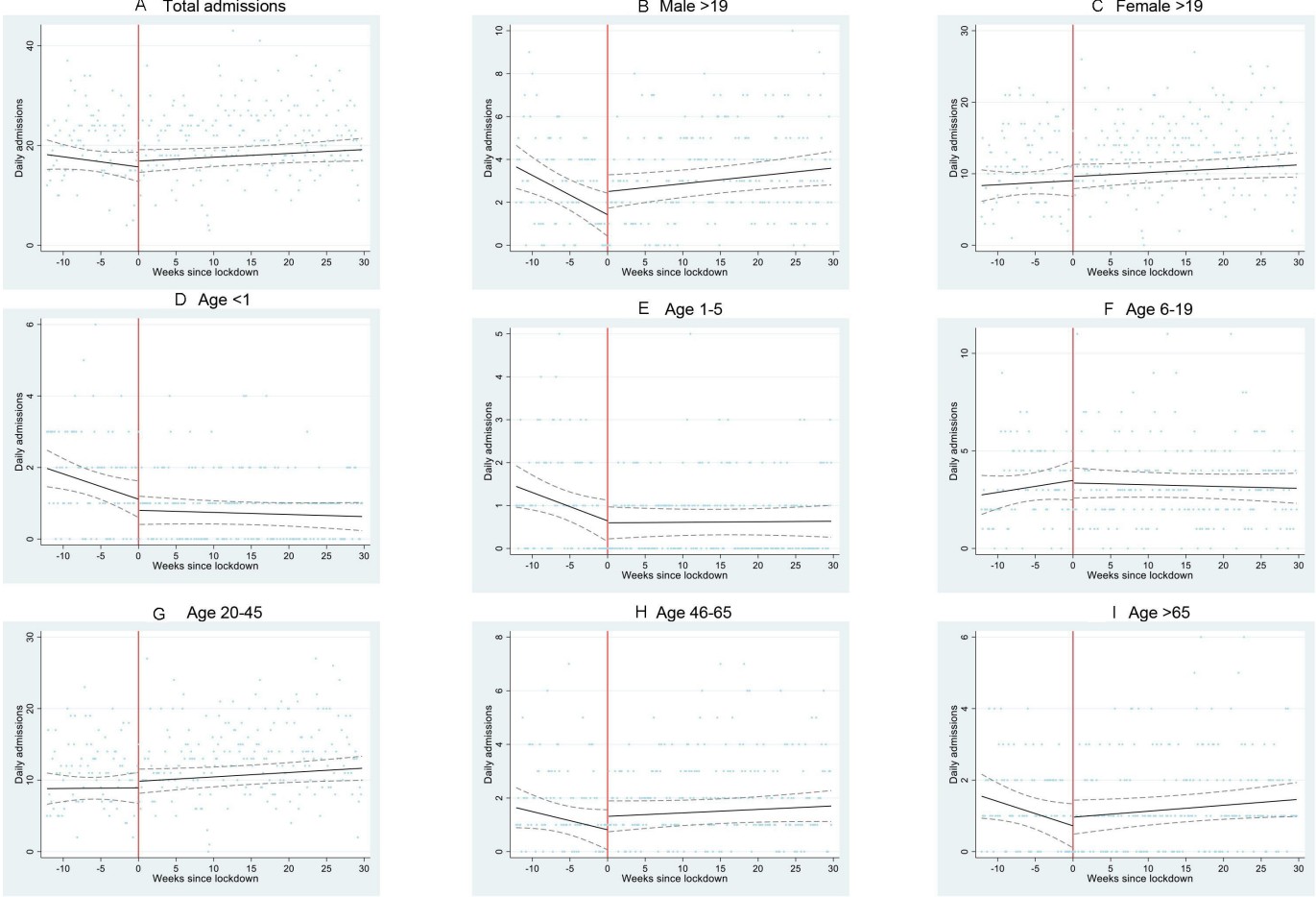

**Figure 2** Admissions to Hlabisa Hospital from January 2020 to October 2020 for all reasons, by sex and age. Scatter plot represents the number of admissions per day for (A) all admissions, (B) men over 19 years old, (C) women over 19 years old, (D) age <1 years old, (E) age 1-5 years old, (F) age 6-19 years old, (G) age 20-45 years old, (H) age 46-65 years old and (I) age >65 years old. Black line represents fitted linear regression model with adjustment for day of week, before and after lockdown. Dashed lines represent CIs of the fitted model. Red line represents the beginning of lockdown level 5.

## DISCUSSION

In this analysis of hospital admission and hospital mortality in a mostly rural South African setting, national lockdown was not associated with reductions in all-cause daily admissions. Total admissions, and of adult men, increased from level 4 to level 3. Numbers of admissions in adult women, individuals over 45 years old and 6–19 years old, and for CDs were mostly unchanged after lockdown. However, admissions for children under 5 fell sharply during level 5 lockdown and remained low. During the final level 1 lockdown, admissions for maternal and neonatal conditions, NCDs and injuries dropped significantly compared with level 2, despite no significant changes during the earlier levels of lockdown. Risk of in-hospital mortality halved during level 5 lockdown then increased to prelockdown rates.

These observed changes could be due to changes in the health of the local population, or to lockdown restrictions, or both. However, the magnitude of changes over a relatively short period and the small number of reported COVID-19 admissions suggest that changes in access to hospital care are likely to have been influential. Reduced

access to hospital during lockdown could be due to a combination of factors including mandatory restrictions on movement, fear of being infected and inability to pay for transport due to lost income (hospital care is free to users). Availability of ambulance transport from home to hospital is limited. Expense of transport and travel time are known barriers to care in this population and are likely to have been exacerbated by restrictions on public transport during lockdown.[27 28]

Our findings that all-cause hospital admissions did not fall but that admissions of children under 5 years old decreased were consistent with findings of Siedner and colleagues[24] on primary care visitations in the same population. However, in the present study under-5 hospital admissions did not increase to normal after level 5, unlike primary care visits which did.[24] Reasons for reduced admissions in under-5s could include reduction in circulating viral illnesses following school closures, or reduction in injuries as children spend more time at home during the strict lockdown measures of level 5. It is unlikely that other seasonal factors such as respiratory viruses or reduction in vitamin D levels are the reasons for

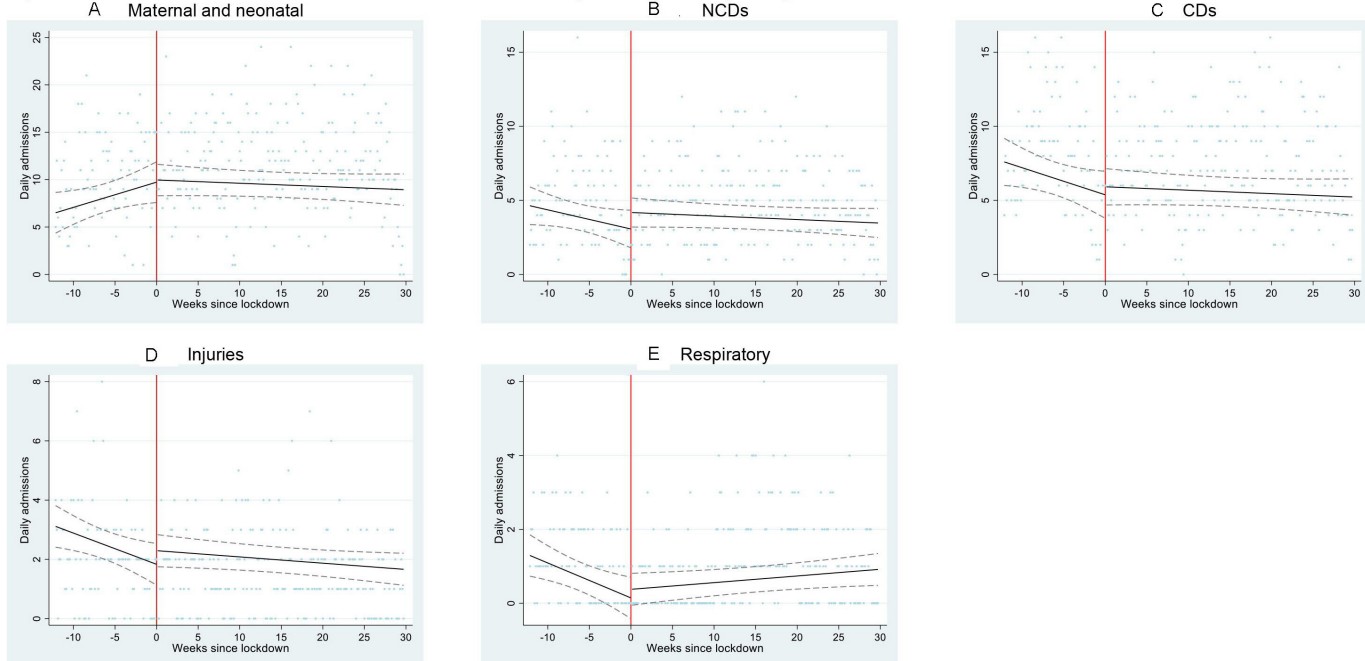

**Figure 3** Admissions to Hlabisa Hospital from January 2020 to October 2020, by discharge diagnosis. Scatter plot represents the number of admissions per day for (A) maternal and neonatal, (B) NCDs, (C) CDs, (D) njuries and (E) respiratory diagnoses Black line represents fitted linear regression model with adjustment for day of week, before and after lockdown. Dashed lines represent CIs of the fitted model. Red line represents the beginning of lockdown level 5. CDs, communicable diseases; NCDs, non-communicable diseases.

the decline in admissions, as lockdown started in the late summer months.[29–31] More concerningly, it could signify that unwell children were not being taken to hospital by caregivers for acute illnesses such as pneumonia or gastroenteritis, among the most common diagnoses in these age groups. This could explain the concurrent drop in respiratory admissions, particularly given that a large proportion of respiratory admissions were in children and most were due to pneumonia (table 1). Reduced access to hospital during level 5 lockdown thus appears to affect children more than other subgroups, highlighting the vulnerability of children and their dependence on carers, who might have been concerned about protecting their young children and themselves from COVID-19 infection.

Similar reductions in paediatric admissions for respiratory disease were reported during the early pandemic in China, and decreased emergency department attendance was reported in Victoria, Australia.[32 33] Modelling by Roberton and colleagues[17] predicted that COVID-19 could cause at least 253 500 additional child deaths in low-income and middle-income countries, 41% of which would be attributable to reduced acute care for childhood illnesses, such as antibiotic treatment of pneumonia. The leading causes of child mortality in this low-income population are acute respiratory infections, HIV-related illness, neonatal pneumonia, diarrhoeal disease and pulmonary tuberculosis, also among the most common reasons for admission to Hlabisa Hospital in under-5s.[34] Reduction in access for treatable illnesses in children may thus have led to preventable mortality during lockdown.

These results suggest that admissions for individuals over 45 years old, CDs and injuries remained stable despite the significant restrictions during lockdown. This is unlike other regions in South Africa which saw reductions in surgical admissions in the North West Province and prenatal antiretroviral pre-exposure prophylaxis visits in the Western Cape.[35 36]

The KwaZulu-Natal provincial health department reported that by 11 April the cumulative incidence of COVID-19 in uMkhanyakude District was 21, with no deaths.[37] By 5 August the cumulative incidence had risen to 1513 cases, accounting for 1.7% of cases in the province. The daily incidence rate was less than 50 cases per 100 000 population.[38] This was despite a peak in incidence of cases nationally in July.[23] Respiratory admissions in this study population (which excluded COVID-19) peaked during level 3, which coincides with the peak in COVID-19 incidence in KwaZulu-Natal. It is possible that undiagnosed COVID-19 presenting as pneumonia could account for some of this increase. The majority of admissions with a primary diagnosis of COVID-19 also occurred during level 3 at Hlabisa, and the combination of both diagnosed and undiagnosed COVID-19 may also partly account for the increase in admissions in men and individuals aged 20–45 years old during level 3 lockdown.

A possible explanation for the increase during level 3 could be migration into the area, if urban workers moved back to their rural homes when transport restrictions eased during level 3 lockdown.[39] However, Siedner and colleagues[24] interrogated local longitudinal primary care

**Table 3** Step changes in odds of death at Hlabisa Hospital at each level of lockdown: logistic regression models*

| | Pre lockdown odds (95% CI) | OR level 5 vs pre lockdown | 95% CI | P value | OR level 4 vs 5 | 95% CI | P value | OR level 3 vs 4 | 95% CI | P value | OR level 2 vs 3 | 95% CI | P value | OR level 1 vs 2 | 95% CI | P value |
|---|---|---|---|---|---|---|---|---|---|---|---|---|---|---|---|---|
| Unadjusted | 0.05 (0.04 to 0.07) | 0.48 | 0.28 to 0.80 | <0.001 | 1.81 | 0.97 to 3.38 | 0.06 | 0.98 | 0.62 to 1.54 | 0.92 | 1.40 | 0.96 to 2.04 | 0.08 | 0.98 | 0.62 to 1.53 | 0.92 |
| Adjusted for day of week, sex and age | 0.02 (0.02 to 0.04) | 0.47 | 0.28 to 0.80 | <0.001 | 1.75 | 0.93 to 3.28 | 0.08 | 0.96 | 0.59 to 1.54 | 0.85 | 1.42 | 0.95 to 2.12 | 0.09 | 0.97 | 0.60 to 1.57 | 0.91 |
| Adjusted for day of week, sex, age and diagnosis | 0.04 (0.03 to 0.07) | 0.48 | 0.28 to 0.83 | 0.01 | 1.94 | 1.01 to 3.71 | 0.05 | 0.84 | 0.51 to 1.39 | 0.51 | 1.74 | 1.15 to 2.63 | 0.01 | 1.00 | 0.59 to 1.71 | 0.99 |

*Analysis at the level of admission, with robust adjustment for intraperson correlation of outcome.

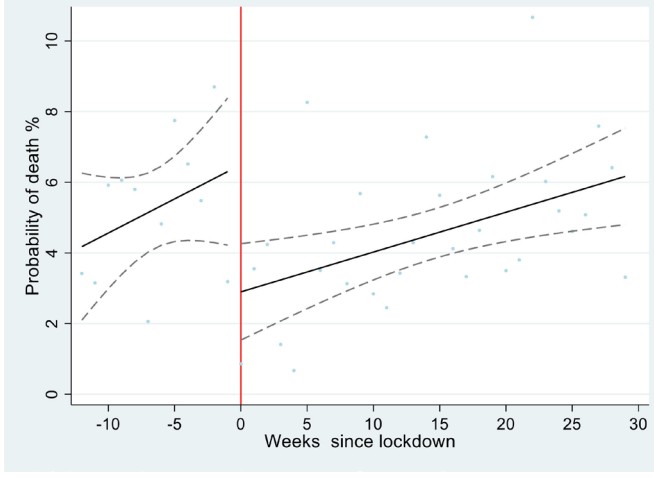

**Figure 4** Probability of death among all admissions each week: observed and predicted by interrupted times series linear regression model. Scatter plot represents probability of death each week. Black line represents fitted linear regression model with adjustment for day of week, before and after lockdown. Dashed lines represent CIs of the fitted model. Red line represents the beginning of lockdown level 5.

attendance data and did not find evidence of substantial migration. The temporary lifting of the ban on alcohol sales at the beginning of level 2 lockdown[20] could have led to an increase in alcohol-related admissions in these sugroups, but admission for injuries or NCDs, which are often alcohol-related, did not change significantly during level 2 lockdown. Admissions for NCDs and injuries also did not decrease significantly at the beginning of the lockdown when the alcohol ban commenced, as seen in Western Cape and Cape Town.[36 40]

A major finding of this study is the large reduction in all-cause mortality during level 5 lockdown, despite resilience in admission numbers overall. This supports our hypothesis that the sickest people may have been disproportionately unable to access hospital care as needed and died at home as opposed to hospital, or presented late to hospital, resulting in increasing mortality from level 4 onwards. Reductions in numbers of admissions in the sickest may have been offset by the observed increases in admissions of younger, healthier adults. The South African Medical Research Council reported 7729 excess deaths in KwaZulu-Natal from 6 May to 10 November 2020, compared with the same dates during previous years, which was much more than the number of COVID-19 cases reported.[41] These excess deaths were probably due both to undiagnosed COVID-19 and to undertreatment of other life-threatening conditions.[41 42] It is thus likely that reduced utilisation by the sickest may have resulted in preventable death outside of hospital—most concerningly this may be in the under-5s as this group was not accessing care for diseases which are a leading cause of mortality in the region. Further analysis of local demographic data on rates and place of death during lockdown may elucidate this finding.

Limitations of this study include only analysing admissions up until 20 October 2020, before the pandemic had resolved globally and the second waves of COVID-19 cases were beginning.[43] Therefore, the impact of the epidemic on long-term healthcare utilisation or hospital mortality cannot yet be assessed in this study. Our analysis is restricted by a relatively small data set from one district hospital, which may have limited the power of our study to detect the effects of lockdown, particularly in subgroup analysis such as in young children and the generalisability to other secondary care settings. While there were no other known significant events during lockdown in the region likely to have influenced utilisation of hospital care, we cannot exclude changes in disease epidemiology and subsequent healthcare demand, weather or sociocultural factors, which may account for the results.

Further work should focus on the impact of COVID-19 on access to children's health services. They appear to have a lower risk of acquiring and suffering from severe COVID-19,[44–46] but the implications of strict lockdowns on young children's access and utilisation of healthcare in low-income settings for acute illnesses are concerning. This study cannot evaluate the reasons for reduced healthcare utilisation in children in this population. However potential solutions may include payback schemes for emergency transport costs (to offset the heightened costs during lockdown). The use of social media may also offer effective means of encouraging parents to seek care at local primary clinics, from where they could be transported to hospital if necessary.

**Acknowledgements** We thank the research assistants, nurses and staff at Hlabisa Hospital and AHRI who contributed towards data collection. We also thank Anita Edwards for proof-reading and support in preparing the manuscript for publication.

**Contributors** MB, MJS, JS and KH conceived the study aims and hypothesis. KH and DG provided AHRI Hospital Information System data. AM and MB designed the study and conducted the statistical analysis, with advice from MJS. AM drafted the manuscript and prepared the tables, figures and final version for publication, with substantial amendment and oversight from MB. All authors discussed the results, critically revised the study and approved the final version of the manuscript.

**Funding** The Africa Health Research Institute is funded by Wellcome Trust (award 201433/Z/16/Z).

**Competing interests** None declared.

**Patient consent for publication** Not required.

**Ethics approval** Ethical approval was granted for AHRI to collect and use data from patients attending the clinics and hospital by the University of KwaZulu-Natal Biomedical Research Ethics Committee (number and title: BE290/16 'A longitudinal population-based platform for epidemiology and intervention research'). This was a secondary analysis on this anonymised data set with permission from AHRI, so further ethical approval was not required.

**Provenance and peer review** Not commissioned; externally peer reviewed.

**Data availability statement** Statistical coding available on reasonable request from corresponding author. AHRI Hospital Information System data are available on request at https://data.ahri.org/index.php/home.

**ORCID iDs**
Amy McIntosh http://orcid.org/0000-0003-3690-2914
Max Bachmann http://orcid.org/0000-0003-1770-3506
Mark J Siedner http://orcid.org/0000-0003-3506-842X
Dickman Gareta http://orcid.org/0000-0002-4004-5655
Janet Seeley http://orcid.org/0000-0002-0583-5272
Kobus Herbst http://orcid.org/0000-0002-5436-9386

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
