## [Reviewer comments · BMJ Open]

ARTICLE DETAILS

TITLE (PROVISIONAL)	Effect of COVID-19 lockdown on hospital admissions and mortality in rural KwaZulu-Natal, South Africa: interrupted time series analysis
AUTHORS	McIntosh, Amy; Bachmann, Max; Siedner, Mark; Gareta, Dickman; Seeley, Janet; Herbst, Kobus

VERSION 1 – REVIEW

REVIEWER	William G. Grant Sunlight, Nutrition and Health Research Center USA
REVIEW RETURNED	18-Dec-2020

GENERAL COMMENTS	This manuscript presents data on hospital admissions and deaths in a rural South African hospital related to lockdowns. Variations were found with reduced hospital deaths after the most severe lockdown. The authors suggest that the lockdown prevented many vulnerable ill people from going to the hospital. That seems like a reasonable hypothesis. Inclusion of 2019 data and adjustment for calendar month did not substantially change any of these results Comment: Could more information regarding this statement be provided? The time period covered in this study extended from 1 January to 20 October 2020. It might be worthwhile to discuss the seasonality of respiratory tract infections as well as 25(OH)D during pregnancy in South Africa. Solar UVB is the primary source of vitamin D. Is there any evidence that during lockdowns in the seasons when vitamin D could be produced from solar UVB that people tended to stay indoors more? On the other hand, since the main purpose of this manuscript seems to be to assess the effect of lockdowns on hospital admission and deaths, the changes in illness and death rates due to changes in solar UVB doses are probably to slow to affect the findings of this study. Thus, most of the following comments and references can probably be ignored other than to use to show that the effect does not play a role in the analysis. Some references to consider citing: The relationship between ultraviolet radiation exposure and vitamin D status. Engelsen O. Nutrients. 2010 May;2(5):482-95. doi: 10.3390/nu2050482.
---

	Winter Is Coming: A Southern Hemisphere Perspective of the Environmental Drivers of SARS-CoV-2 and the Potential Seasonality of COVID-19. Smit AJ, Fitchett JM, Engelbrecht FA, Scholes RJ, Dzhivhuho G, Sweijid NA. Int J Environ Res Public Health. 2020 Aug 5;17(16):5634. doi: 10.3390/ijerph17165634. Global dynamic spatiotemporal pattern of seasonal influenza since 2009 influenza pandemic. Xu ZW, Li ZJ, Hu WB. Infect Dis Poverty. 2020 Jan 3;9(1):2. doi: 10.1186/s40249-019-0618-5. Heterogeneity in influenza seasonality and vaccine effectiveness in Australia, Chile, New Zealand and South Africa: early estimates of the 2019 influenza season. Sullivan SG, Arriola CS, Bocacao J, Burgos P, Bustos P, Carville KS, Cheng AC, Chilver MB, Cohen C, Deng YM, El Omeiri N, Fasce RA, Hellferscee O, Huang QS, Gonzalez C, Jelley L, Leung VK, Lopez L, McAnerney JM, McNeill A, Olivares MF, Peck H, Sotomayor V, Tempia S, Vergara N, von Gottberg A, Walaza S, Wood T. Euro Surveill. 2019 Nov;24(45):1900645. doi: 10.2807/1560-7917.ES.2019.24.45.1900645. Global patterns in monthly activity of influenza virus, respiratory syncytial virus, parainfluenza virus, and metapneumovirus: a systematic analysis. Li Y, Reeves RM, Wang X, Bassat Q, Brooks WA, Cohen C, Moore DP, Nunes M, Rath B, Campbell H, Nair H; RSV Global Epidemiology Network; RESCEU investigators. Lancet Glob Health. 2019 Aug;7(8):e1031-e1045. doi: 10.1016/S2214-109X(19)30264-5. Decreased Influenza Activity During the COVID-19 Pandemic - United States, Australia, Chile, and South Africa, 2020. Olsen SJ, Azziz-Baumgartner E, Budd AP, Brammer L, Sullivan S, Pineda RF, Cohen C, Fry AM. MMWR Morb Mortal Wkly Rep. 2020 Sep 18;69(37):1305-1309. doi: 10.15585/mmwr.mm6937a6. Maternal and neonatal vitamin D status at birth in black South Africans. Velaphi SC, Izu A, Madhi SA, Pettifor JM. S Afr Med J. 2019 Sep 30;109(10):807-813. doi: 10.7196/SAMJ.2019.v109i10.13651. Maternal plasma vitamin D levels and associated determinants in late pregnancy in Harare, Zimbabwe: a cross-sectional study. Chikwati RP, Musarurwa C, Duri K, Mhandire K, Snyman T, George JA. BMC Pregnancy Childbirth. 2019 Jun 28;19(1):218. doi: 10.1186/s12884-019-2362-z.
--	---

REVIEWER	Luis Puente Maestu Hospital Universitario Gregorio Marañón- Universidad Complutense de Madrid Spain
REVIEW RETURNED	31-Dec-2020

GENERAL COMMENTS	This study describes the impact on admission to a hospital serving a rural area of south-Africa. The authors do not find differences except for children younger than 5 years. In the strictest period of lock down there was also a reduction in in-hospital mortality. The paper describes the experience that, while interesting in itself, is
---

	hardly translatable to other settings since the number of cases of respiratory disease (among them COVID-19, I guess) was very low and the age and reason for admissions in quite different compared with other settings. This reader also would anticipate less terror among the population attended at the Hablista hospital than in Europe and North-America and, as far as the authors comment, from other areas of South Africa as well. 1. The paper is long and difficult to follow. The authors may see to reduce it. The elegance of a text rests on using appropriate words and reducing the number of words as much as possible without losing the meaning. I make some suggestion in the specific comments, but there are many more instances in which the authors can trim the text. 2. Do the authors have information about the number of deaths outside the hospital in their area? 3. Page 4, line 38. The description of the low-down phases is too detailed. It can be said just that the different stages compelled to different levels of movement and opening of business restrictions and describe them in the supplementary on-line material 4. Page 6 line 19 We first calculated the number of admissions each day then analysed the aggregated data at day level. These analyses were carried out for all admissions from 1st January 2020 to 20th October 2020, and then repeated in subgroups defined by age, sex and diagnosis. This information can be omitted 5. Page 6 line 2 "To estimate absolute changes in mean numbers" 6. Page 6 line 29 "To estimate absolute changes in mean numbers" can be said to estimate mean changes or changes in the variables mean 7. Page 6 line 36 "We calculated the number of admissions each day". Calculated or measured?
--	---

REVIEWER	VINCENT HOOPER XIAMEN UNIVERSITY, MALAYSIA
REVIEW RETURNED	02-Jan-2021

GENERAL COMMENTS	AN EXCELLENT STUDY CONDUCTED BY VERY WELL ESTABLISHED SCHOLARS AT LEADING INSTITUTIONS WORTHY OF FAST PUBLICATION IN BMJ OPEN.
--

REVIEWER	Vasco Ricoca Peixoto NOVA National School of Public Health, Public Health Research Centre, Universidade NOVA de Lisboa
REVIEW RETURNED	05-Jan-2021

GENERAL COMMENTS	It is a good work and a good read with relevant original, context specific findings. Title and abstract - Make sure it is clear from title and objective that the outcome is "All cause admissions" Abstract - Methods could be further clarified in Abstract. Introduction - Consider a visual aid such as table or incidence graph with moments of each lockdown level initiation and respective main measures.
---

	Results Table 2. Consider changing top " IRR at level 5 - (Ref:Level 4)" Also not clear why level 5 vs 4 and in the next column 4 vs 5. Consider comparing pre lockdown as reference against all lockdown levels to see changes in IRR and OR for respective outcomes in a more clear way f. Discussion - Are there any other reference or data for Regional mortality?How did it change for those younger than 5? Is it reported? Can people easily activate emergency transportation? Do you expect that this potential illness that would have been admitted were severe enough to cause death or disability in your specific context? A brief reference to general standard of living in this rural area is relevant for the reader to grasp risk of acute illness in younger than 5 that should have been admitted and were not. What about infectious gastrointestinal illness with dehydration in <5. Is respiratory sincithial virus relevant cause of bronchiolitis? One limitation that is refered (small dataset) seems particularly relevant for those younger than 5. What absolut number of admissions were prevented by the lockdown in that age group during the whole period? Could be interesting to discuss. We may infer by the daily reduction to half and consider the period but its interesting for discussion. but the implications of strict lockdowns on young children’s access and utilisation to healthcare for acute illnesses can be concerning in midle and low income countries. End with reccomendations for similar socio-economic contexts further than emmergency transport (the background of emmergency transport should be previously discussed). What would be needed ? Information to make people seek emmergent healthcare when appropriate? Health phonelines to assess symptom severuty and need for transportt? lack of transport? adress fear of hospital/COVID?
--	---

VERSION 1 – AUTHOR RESPONSE

Reviewer 1: Mr William Grant

1. "Inclusion of 2019 data and adjustment for calendar month did not substantially change any of these results: Could more information regarding this statement be provided?",

Response: We have included more detail regarding this sensitivity analysis in the results section: the full results of this analysis were not included the final manuscript as we felt it may overestimate the effect size by using all of the 2019 data.

2. "It might be worthwhile to discuss the seasonality of respiratory tract infections as well as 25(OH)D during pregnancy in South Africa".

Response: We have discussed how this is unlikely to have influenced the results given lockdown started in the late summer months in the discussion and included some of the suggested references.

Reviewer 2: Dr Luis Puente-Maestu

3. "The paper is long and difficult to follow. The authors may see to reduce it. "

Response: We have revised the paper throughout to improve clarity and concision.

4. "Do the authors have information about the number of deaths outside the hospital in their area?"

Response: We do not have access to these data for the study period. However, in the results section, we have provided additional information about mortality trends in the province at the time.

5. "Page 4, line 38. The description of the low-down phases is too detailed. It can be said just that the different stages compelled to different levels of movement and opening of business restrictions and describe them in the supplementary on-line material."

Response: We believe it is important to describe what each level entailed, because the study compares each level of lockdown which appeared to have different effects on hospital admissions. However, to take these details out of the text we have put them in Figure 1.

6. "Page 6 line 19 We first calculated the number of admissions each day then analysed the aggregated data at day level. These analyses were carried out for all admissions from 1st January 2020 to 20th October 2020, and then repeated in subgroups defined by age, sex, and diagnosis. This information can be omitted."

Response: We would prefer to keep this methodological information which defines how admissions were calculated and analysed (at day level), which is different from the analysis of mortality. However, we have shortened the second sentence because the dates were stated earlier.

7. "Page 6 line 2 "To estimate absolute changes in mean numbers" and "Page 6 line 29 "To estimate absolute changes in mean numbers" can be said to estimate mean changes or changes in the variables mean"

Response: We use the wording "absolute changes" to specify that these estimates were differences in means, in contrast to incident rate ratios which are relative changes.

8. "Page 6 line 36 "We calculated the number of admissions each day". Calculated or measured?"

Response: We have omitted this sentence as this is described in detail elsewhere with more clarity.

Reviewer 4: Dr. VASCO RICOCA FREIRE DUARTE Peixoto

9. "Title and abstract - Make sure it is clear from title and objective that the outcome is All cause admissions".

Response: We have changed the objective to "To assess the effect of lockdown during the 2020 COVID-19 pandemic on daily all- cause admissions, and by age and diagnosis sub-groups, and odds of all-cause mortality..." However we felt that changing the title to "all cause admissions" would be misleading because many of the results are about admissions for specific diseases or age groups.

10. "Abstract - Methods could be further clarified in Abstract"

Response: We have tried to make these as clear as possible while following the journal's format.

11. "Introduction - Consider a visual aid such as table or incidence graph with moments of each lockdown level initiation and respective main measures."

Response: As stated above, we have put this information in Figure 1.

12. "Results Table 2. Consider changing top " IRR at level 5 - (Ref:Level 4)"

Also not clear why level 5 vs 4 and in the next column 4 vs 5"

Response: We have corrected this column heading in Table 2.

13. "Consider comparing pre lockdown as reference against all lockdown levels to see changes in IRR and OR for respective outcomes in a more clear way."

Response: In our analysis we did compare each lockdown level with pre lockdown, as well as comparing successive lockdown levels which we report in the Tables. We reported only the latter results because reporting both would double the amount of information in the Tables 2 and 3. The incidence rate ratios and odds ratios comparing each lockdown level with pre lockdown can be

calculated from the tables by multiplying the relevant IRRs or ORs. However, in response to this comment we have added asterixis to Table 2, and included this in the results, indicating where there were statistically significantly differences from pre-lockdown (these were not applicable to Table 3 because only level 5 was significantly different from pre-lockdown, which is already shown).

14. "Are there any other reference or data for Regional mortality? How did it change for those younger than 5? Is it reported?"

Response: As stated in response to reviewer 2, we have provided additional information on mortality in the province the Discussion: "The South African Medical Research Council reported 7729 excess deaths in KwaZulu-Natal from 6th May to 10th November 2020, compared to the same dates during previous years, which was much more than the number of COVID-19 cases reported [41]."

15. "Can people easily activate emergency transportation?"

Response: We have added to Discussion: "Availability of ambulance transport from home to hospital is limited. Expense of transport and travel time are known barriers to care in this population, and are likely to have been exacerbated by restrictions on public transport during lockdown."

16. "Do you expect that this potential illness that would have been admitted were severe enough to cause death or disability in your specific context? A brief reference to general standard of living in this rural area is relevant for the reader to grasp risk of acute illness in younger than 5 that should have been admitted and were not."

Response: We have added to Discussion: "The leading causes of child mortality in this low-income population are acute respiratory infections, HIV-related illness, neonatal pneumonia, diarrhoeal disease and pulmonary TB: also amongst the most common reasons for admission to Hlabisa hospital in under fives.[34] Reduction in access for treatable illnesses in children may thus have led to preventable mortality during lockdown. "

17. "What about infectious gastrointestinal illness with dehydration in <5. Is respiratory syncytial virus relevant cause of bronchiolitis?"

Response: We have added to Discussion: "It is unlikely that other seasonal factors such as respiratory viruses or reduction in Vitamin D production is the reason for the decline in admissions as lockdown started in the late summer months [29–31]. More concerningly, it could signify that unwell children were not being taken to hospital by caregivers for acute illnesses such as pneumonia or gastroenteritis, among the most common diagnoses in these age groups."

18. "One limitation that is referred (small dataset) seems particularly relevant for those younger than 5.

What absolute number of admissions were prevented by the lockdown in that age group during the whole period? Could be interesting to discuss. We may infer by the daily reduction to half and consider the period but its interesting for discussion. But the implications of strict lockdowns on young children's access and utilisation to healthcare for acute illnesses can be concerning in middle and low income countries."

Response: In the text of Results we have provided additional information about absolute changes in daily admissions in under fives: "In infants this represents a reduction in the incidence rate from 1.46 to 0.91 admissions per day, which . Overall, the incidence rate did not recover to the baseline by the end of the study, with: incidence rate 0.42 admissions per day during level 1 lockdown. The same trend was seen in 1-5 year olds, with: incidence rate before lockdown was 1.07 per day⁶, reducing to 0.43 admissions per day during level 5, without and not recovering to baseline by level 1, with incidence rate of 0.61 per day."

Additional response: The study does not provide evidence about the reasons why children were unable to access care, but one possibility is difficulty in accessing transport, and so we have included more details on the transport in the region and the lockdown implications of this. Other

recommendations are only theoretical given we do not know the exact cause of why children were not accessing care.

19. “End with recommendations for similar socio-economic contexts further than emergency transport (the background of emergency transport should be previously discussed). What would be needed ? Information to make people seek emergency healthcare when appropriate? Health phonelines to assess symptom severity and need for transport? lack of transport? address fear of hospital/COVID? “

Response: We have added suggestions for improving access to hospital at the end of Discussion.

VERSION 2 – REVIEW

REVIEWER	William B. Grant Sunlight, Nutrition and Health Research Center USA
REVIEW RETURNED	01-Feb-2021
GENERAL COMMENTS	Thanks for making the changes. No further comments.
REVIEWER	Luis Puente Universidad Complutense Medical School- Medicien Department
REVIEW RETURNED	07-Feb-2021
GENERAL COMMENTS	The reviewer completed the relevant checklist but made no further comments.
REVIEWER	Vasco Ricoca Peixoto NOVA National School of Public Health
REVIEW RETURNED	22-Feb-2021
GENERAL COMMENTS	Comments in previous review generally adressed. Some english could be made clearer, more gramaticaly correct.